# Vision-Language Navigation with Energy-Based Policy

**Rui Liu**[1,2]     **Wenguan Wang**[2]     **Yi Yang**[1,2,*]

[1]The State Key Lab of Brain-Machine Intelligence, Zhejiang University, Hangzhou, China
[2]College of Computer Science and Technology, Zhejiang University, Hangzhou, China
`https://github.com/DefaultRui/VLN-ENP`

## Abstract

Vision-language navigation (VLN) requires an agent to execute actions following human instructions. Existing VLN models are optimized through expert demonstrations by supervised behavioural cloning or incorporating manual reward engineering. While straightforward, these efforts overlook the accumulation of errors in the Markov decision process, and struggle to match the distribution of the expert policy. Going beyond this, we propose an Energy-based Navigation Policy (ENP) to model the joint state-action distribution using an energy-based model. At each step, low energy values correspond to the state-action pairs that the expert is most likely to perform, and vice versa. Theoretically, the optimization objective is equivalent to minimizing the forward divergence between the occupancy measure of the expert and ours. Consequently, ENP learns to globally align with the expert policy by maximizing the likelihood of the actions and modeling the dynamics of the navigation states in a collaborative manner. With a variety of VLN architectures, ENP achieves promising performances on R2R, REVERIE, RxR, and R2R-CE, unleashing the power of existing VLN models.

## 1 Introduction

Vision-language navigation (VLN) [1] entails an embodied agent that executes a sequence of actions to traverse complex environments based on natural language instructions. VLN is typically considered as a Partially Observable Markov Decision Process (POMDP) [2], where future states are determined only by the current state and current action without explicit rewards. Therefore, the central challenge for VLN is to learn an effective navigation policy. In this context, existing neural agents [1, 3, 4, 5, 6] are naturally constructed as data-driven policy networks by imitation learning (IL) [7] and bootstrapping from expert demonstrations, *i.e.*, ground-truth trajectories [1].

Most VLN models [1, 3, 8, 4, 5, 6] utilize behavioural cloning (BC), a classic approach for IL, through supervised training. While conceptually simple, BC suffers seriously from quadratic accumulation errors [9, 10] along the trajectory due to covariate shift, especially in partially observable settings. Several efforts introduce 'student-forcing' [1, 11] and 'pseudo interactive demonstrator' [6], which are essentially online versions of DAGGER [12], to alleviate this distribution mismatch. DAGGER assumes interactive access to expert policies (*i.e.*, the shortest path from current state to the goal[2]) and reduces to linear expected errors [12, 10]. Some other studies [13, 3, 8, 4, 5] combine reinforcement learning (RL) [14] and IL algorithms to mitigate the limitations of BC. Though effective, it presents challenges in designing an optimal reward function [15] and demands careful manual reward engineering. Such reward functions may not be robust to changes in the environment dynamics and

---

[*]Corresponding author: Yi Yang.

[2]These approaches require a slightly more interactive setting than traditional imitation learning, allowing the learner to query the expert at any navigation state during training. Previous studies [1, 6] have adopted this setting, and this assume is feasible for many real-world imitation learning tasks [12].

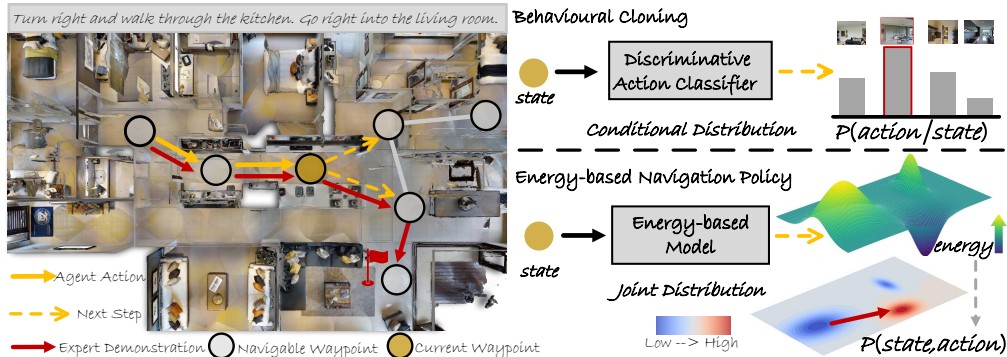

Figure 1: Comparison of behavioural cloning (BC) and ENP for VLN. Previous methods use BC to optimize the conditional action distribution directly. ENP models the joint state-action distribution through an energy-based model. The low energy values correspond to the state-action pairs that the expert is most likely to perform.

struggle to generalize well across diverse scenes [16]. In addition, recent research [6] has revealed that training transformer [17]-based VLN models using RL would introduce instability.

From a probabilistic view, most VLN agents [1, 3, 4, 6] essentially train a discriminative action classifier [8, 3, 4], and model the action distribution conditioned on current navigation state $P(a|s)$ at each time step (Fig. 1). This classifier is supervised by the cross-entropy loss to minimize 1-step deviation error of single-step decision along trajectories [18]. However, the prediction of current action affects future states during the global execution of the learned policy, potentially leading to catastrophic failures [9, 12]. In other words, solely optimizing the conditional action probabilities (*i.e.*, policy) remains unclear for the underlying reasons of the expert behaviour, yet fails to exploit complete information in the state-action distribution $P(s, a)$ induced by the expert [14].

In light of this, we propose an Energy-based Navigation Policy (ENP) framework that addresses the limitations of current policy learning in VLN from an energy perspective (Fig. 1). Energy-based models (EBMs) [19] are flexible and capable of modeling expressive distributions, as they impose minimal restrictions on the tractability of the normalizing constant (partition function) [20]. We reinterpret the VLN model as an energy-based model to learn the expert policy from the expert demonstrations. Specifically, ENP models the joint distribution of state-action pairs, *i.e.*, $P(s, a)$, by energy function, and assigns low energy values to the state-action pairs that the expert mostly performs. For optimization, this energy function is regarded as the unnormalized $\log$-density of joint distribution, which can be decomposed into the standard cross-entropy of action prediction and marginal distribution of the states $P(s)$. During training, ENP introduces persistent contrastive divergence [21, 22] to estimate the expectation of $P(s)$, and samples by Stochastic Gradient Langevin Dynamics [23]. In this way, ENP simulates the expert by maximizing the likelihood of the action and incorporates prior knowledge of VLN by optimizing state marginal distribution in a collaborative manner, thereby leveraging the benefits of both energy-based and discriminative approaches.

Theoretically, the training objective of ENP is to maximize the expected $\log$-likelihood function over the joint distribution $P(s, a)$. This is equivalent to estimating the unnormalized probability density (*i.e.*, energy) of the expert's occupancy measure [15, 24, 25]. Therefore, ENP performs prioritized optimization for entire trajectories rather than the single-time step decisions in BC, and achieves a global alignment with the expert policy. Furthermore, we realize that ENP is closely related to Inverse Reinforcement Learning (IRL) [7, 26, 15]. For the optimized objective, ENP shares similarities with IRL but minimizes the forward KL divergence [27, 28] between the occupancy measures.

For thorough examination (§4), we explore ENP across several representative VLN architectures [3, 4, 6, 29]. Experimental results demonstrate that ENP outperforms the counterparts, *e.g.*, **2**% SR and **1**% SPL gains over R2R [1], **1.22**% RGS on REVERIE [30], **2**% SR on R2R-CE [31], and **1.07**% NDTW on RxR-CE [32], respectively. ENP not only helps release the power of existing VLN models, but also evidences the merits of energy-based approaches in the challenging VLN decision-making.

## 2 Related Work

**Vision-Language Navigation (VLN).** Early VLN agents [1, 13, 3, 33] are built upon a recurrent neural policy network within a sequence-to-sequence framework [34, 35] to predict action distribution.

As compressing all history into a hidden vector might be sub-optimal for state tracking across extended trajectories, later attempts [36, 37, 6, 38] incorporate a memory buffer (*e.g.*, topological graph) for long-term planning. Recent efforts [39, 5, 4] are devoted to encode complete episode history of navigation states and actions by transformer [17] and optimize the whole model in end-to-end training. Later, various strategies have been proposed to improve the generalization of policies in both seen and unseen environments [3, 8, 40, 41], such as progress estimation [42], instruction generation [33, 43, 44, 45, 46], backtracking [47], cross-modal matching [48, 49], self-supervised pre-training [50, 51, 52], environmental augmentation [3, 53, 54, 55], visual landmarks [56, 57], exploration [58, 59], map building [60, 61, 62, 63], waypoint prediction [64], and foundation models [65, 66].

**Policy Learning in VLN.** VLN can be viewed as a *POMDP* [2], providing several expert demonstrations sourced from ground-truth shortest-path trajectories [1, 30] or human demonstrations [32]. Behavioural cloning (BC) [67, 68], a typical approach of imitation learning (IL) [7, 15], is widely used in current VLN agents [1, 33, 4, 6] with supervised policy learning. Nevertheless, it suffers from distribution shifts between training and testing [12, 69]. [1] introduces 'student-forcing' [11] training regime from sequence prediction to mitigate this limitation. Later agents [3, 8, 4, 5] combine both IL and model-free RL [14, 70] for policy learning, where the reward function is manually designed on the distance metric. Instead of directly mapping visual observations into actions or state-action values, [13, 71] investigate model-based RL [72, 73] for look-ahead planning, exploring the potential future states. As reward engineering requires careful manual tuning, [16] proposes to learn reward functions from the expert distribution directly. Although RL is an effective approach in principle, DUET [6] finds it difficult to train large-scale transformers with inaccurate and sparse RL rewards [74] in VLN. Existing studies [61, 62, 63] use an interactive demonstrator (similar to DAGGER [12]) to generate additional pseudo demonstrations and achieve promising performance.

In this paper, ENP learns to align with the expert policy by matching the joint state-action distribution for entire trajectories from the occupancy measure view. Furthermore, ENP reinterprets the optimization of policy into an energy-based model, representing an expressive probability distribution.

**Energy-based Model.** Energy-based models (EBMs) [19] are non-normalized probabilistic models that represent data using a Boltzmann distribution. In EBMs, each sample is associated with an energy value, where high-density regions correspond to lower energy [75]. A range of MCMC-based [76, 21] and MCMC-free [77, 78] approaches can be adopted to estimate the gradient of the log-likelihood. EBMs have been widely utilized in generation [22], perception [79], and policy learning [80, 81]. The most related work [25, 24, 22] will be discussed below. [25] employs EBM for distribution matching, strictly prohibiting interaction with the environment. Moreover, this approach is applied in an offline setting, specifically for healthcare. [24] considers the expert energy as the reward function and leverages this reward for reinforcement learning. [22] adopts parameterized energy functions to model the conditional action distribution, optimizing for actions that minimize the energy landscape.

In contrast, ENP devotes to learning the joint distribution of state-action pairs from the expert demonstrations online. An external marginal state memory is introduced to store the historical information for initializing the samples for next iterations. ENP demonstrates the potential of EBMs in VLN decision-making, and improves the existing VLN agents across various frameworks.

# 3 Method

In this section, we first formalize the VLN task and discuss the limitations of existing work from a probabilistic view (§3.1). Then we introduce Energy-based Navigation Policy learning framework – ENP (§3.2). Furthermore, ENP is compared with other imitation learning methods based on divergence minimization (§3.3). Finally, we provide the implementation details (§3.4).

## 3.1 Problem Formulation

VLN is typically formulated as a POMDP [2, 51] with a finite horizon $T$, defined as the tuple $<\mathcal{S}, \mathcal{A}, \mathcal{T}, \mathcal{R}, \rho_0>$. $\mathcal{S}$ is the navigation state space. $\mathcal{A}$ is the action space. Note that we use panoramic action space [33], which encodes high-level behaviour. The transition distribution $\mathcal{T}(s_{t+1}|s_t, a_t)$ (Markov property) of the environment, the reward function $\mathcal{R}(s_t, a_t)$, and the initial state distribution $\rho_0$ are unknown in VLN and $\mathcal{T}$ can only be queried through interaction with the environment.

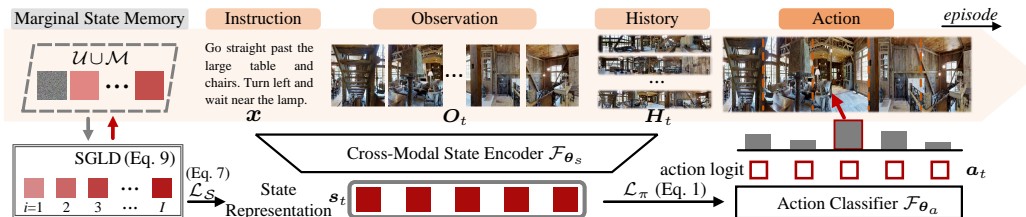

Figure 2: Overview of ENP. At each step $t$, the agent acquires a series of observations, and predicts the next step based on the instruction and navigation history. ENP optimizes the marginal state matching loss $\mathcal{L}_S$ through SGLD sampling from Marginal State Memory (Eq. 9), and minimizes the cross-entropy loss $\mathcal{L}_\pi$ jointly (Eq. 1).

In the standard VLN setting [1], a training dataset $\mathcal{D} = \{(x, \tau)\}$ consists of pairs of the language instructions $x$ and corresponding expert trajectories $\tau$. At step $t$, the agent acquires an egocentric observation $O_t$, and then takes an action $a_t \in \mathcal{A}$ from $K_t$ candidates based on the instruction $x$. The agent needs to learn a navigation policy $\pi^*(a|s)$ and predict the action distribution $P(a_t|s_t)$.

Recent VLN agents employ Seq2Seq [8, 3] or transformer-based networks [4, 6] as a combination of a cross-modal state encoder $\mathcal{F}_{\boldsymbol{\theta}_s}$ and a conditional action classifier $\mathcal{F}_{\boldsymbol{\theta}_a}$ for action prediction. Hence they are usually built as a composition of $\mathcal{F}_{\boldsymbol{\theta}_s} \diamond \mathcal{F}_{\boldsymbol{\theta}_a}$. There are different state representations in previous work [51, 4], here we define a more general one. The cross-modal state encoder extracts a joint state representation $s_t = \mathcal{F}_{\boldsymbol{\theta}_s}(x, O_t, H_t) \in \mathbb{R}^{K_t \times D}$ by the visual embedding $O_t$, the instruction embedding $x$, and history embedding $H_t$ along the trajectory $\tau$. Concretely, $H_t$ comes from recurrent hidden units of Seq2Seq framework or an external memory module (*e.g.*, topological graph [6]). Then, the action classifier maps the state representation $s_t$ into an action-related vector $a_t = \{a_t[k]\}_{k=1}^{K_t} \in \mathbb{R}^{K_t}$, *i.e.*, $a_t = \mathcal{F}_{\boldsymbol{\theta}_a}(s_t)$, and $a_t[k]$ is termed as the logit for $k$-th action candidate. Then the probability of $a_{t,k}$ is computed by softmax, *i.e.*, $P(a_{t,k}|s_t) = \frac{\exp(a_t[k])}{\sum_{k=1}^{K} \exp(a_t[k])}$.

The cross-modal state encoder $\mathcal{F}_{\boldsymbol{\theta}_s}$ and action classifier $\mathcal{F}_{\boldsymbol{\theta}_a}$ usually employ a joint end-to-end training. Imitation learning (IL) aims to learn the policy from expert demonstrations $\mathcal{D}$ without any reward signals. Behaviour cloning (BC) [68] is a straightforward approach of IL by maximizing likelihood estimation of $\pi^*(a|s)$, *i.e.*, minimizing the cross-entropy loss in a supervised way:

$$\arg\min_{\boldsymbol{\theta}} -\mathbb{E}[\log \pi^*(a|s)] \to \mathcal{L}_\pi = \sum\nolimits_{(x,\tau)\sim\mathcal{D}} \sum\nolimits_t - \log P_{\boldsymbol{\theta}}(a_t|s_t), \tag{1}$$

where $\boldsymbol{\theta} = \boldsymbol{\theta}_s \cup \boldsymbol{\theta}_a$. BC suffers from the covariate shift problem for the crucial *i.i.d.* assumption, as the execution of $a_t$ will affect the distribution of future state $s_{t+1}$ [9]. Inspired by DAGGER [12], some agents [1, 6] query the shortest paths (by expert policy $\pi^e$) during navigation, explore more trajectories $\{\tau^+\}$ by interacting with the environments, and aggregate these trajectories as $\mathcal{D}^+ = \{(x, \tau)\} \cup \{(x, \tau^+)\}$. Then, the policy is updated by mixing iterative learning [12]. However, these traditional approaches ignore the changes in distribution and train a policy that performs well under the distribution of states encountered by the expert. When the agent navigates in the unseen environment and receives different observations than expert demonstrations, it will lead to a compounding of errors, as they provide no way to understand the underlying reasons for the expert behaviour [27].

## 3.2 Energy-based Expert Policy Learning

Instead of solely training the discriminative action classifier at each step, we want to learn a policy $\pi^*$ by exploiting the complete information in the state-action distribution from the expert policy $\pi^e$. To facilitate later analysis, the occupancy measure $\rho^\pi(s, a)$ of the policy $\pi$ is introduced [14]. $\rho^\pi(s, a)$ denotes the stationary distribution of state-action pairs $(s, a)$ when an agent navigates with policy $\pi$ in the environment, defined as:

$$\rho^\pi(s, a) \triangleq \pi(a|s) \sum\nolimits_t P(s_t = s|\pi) = \sum\nolimits_t P(s_t = s, a_t = a), \tag{2}$$

and $\rho^\pi(s, a)$ has a one-to-one relationship with policy $\pi$: $\rho^\pi = \rho^{\pi^*} \Leftrightarrow \pi = \pi^*$ [15]. Therefore, we induce a policy $\pi^*$ with $\rho^*(s, a)$ that matches the expert $\rho^e(s, a)$ based on the joint distribution of state-action pairs $P(s, a)$. $t$ is omitted for brevity. The matching loss $\mathcal{L}_\rho$ is designed as:

$$\arg\min -\mathbb{E}_{(s,a)\sim\rho^e(s,a)}[\log \rho^*(s, a)] \propto \mathcal{L}_\rho = -\sum\nolimits_{(x,\tau)\sim\mathcal{D}} \log P_{\boldsymbol{\theta}}(s, a). \tag{3}$$

Then we reformulate it from an energy view, as energy-based models (EBMs) [19, 79] parameterize an unnormalized data density and can represent a more expressive probability distribution. Specifically,

EBMs are first introduced to model $P(s, a)$ of the agent as a Boltzmann distribution:

$$P_{\boldsymbol{\theta}}(s, a) = \frac{\exp(-E_{\boldsymbol{\theta}}(s, a))}{Z_{\boldsymbol{\theta}}}, \tag{4}$$

where $E_{\boldsymbol{\theta}}(s, a) : \mathcal{S} \times \mathcal{A} \to \mathbb{R}$ is the energy function, and $Z_{\boldsymbol{\theta}}$ is the partition function (normalizing constant). As it is difficult to optimize the joint distribution directly [79], we factorize it as (Fig. 2):

$$-\log P_{\boldsymbol{\theta}}(s, a) = -\log P_{\boldsymbol{\theta}}(a|s) - \log P_{\boldsymbol{\theta}}(s) \to \mathcal{L}_{\rho} = \mathcal{L}_{\pi} + \mathcal{L}_{\mathcal{S}}. \tag{5}$$

The first term, $\log P(a|s)$, is typically from a discriminative classifier and can be optimized by minimizing the cross-entropy loss $\mathcal{L}_{\pi}$ as in Eq. 1. As stated in [82, 83], the marginal state distribution matching loss $\mathcal{L}_{\mathcal{S}}$ provides an explicit objective for navigation exploration and incorporates prior knowledge from the expert demonstrations. $P(s)$ is computed by marginalizing out $a$ as:

$$P_{\boldsymbol{\theta}}(s) = \sum_{k=1}^{K} P_{\boldsymbol{\theta}}(s, a_k) = \frac{\sum_{k=1}^{K} \exp(-E_{\boldsymbol{\theta}}(s, a_k))}{Z_{\boldsymbol{\theta}}}. \tag{6}$$

It is worth noting that $\log P(s)$ is the likelihood of an EBM, which cannot be computed directly due to the intractable partition function $Z_{\boldsymbol{\theta}}$. A common approach is to estimate the gradient of the log-likelihood for likelihood maximization with gradient ascent [75]:

$$\begin{aligned} \nabla_{\boldsymbol{\theta}} \mathcal{L}_{\mathcal{S}} \to -\nabla_{\boldsymbol{\theta}} \log P_{\boldsymbol{\theta}}(s) &= -\nabla_{\boldsymbol{\theta}} \log \sum_{k=1}^{K} \exp(-E_{\boldsymbol{\theta}}(s, a_k)) + \nabla_{\boldsymbol{\theta}} \log Z_{\boldsymbol{\theta}} \\ &= \nabla_{\boldsymbol{\theta}} E_{\boldsymbol{\theta}}(s) - \mathbb{E}_{\hat{s} \sim P_{\boldsymbol{\theta}}(\hat{s})}[\nabla_{\boldsymbol{\theta}} E_{\boldsymbol{\theta}}(\hat{s})]. \end{aligned} \tag{7}$$

Considering the low energy value corresponds to the state-action pairs that the expert mostly perform, the action logits $\boldsymbol{a}_t[k]$ of the agent can be used to describe such energy value as $E_{\boldsymbol{\theta}}(s_t, a_{t,k}) \triangleq -a_{t,k}$ formally. Then, the first gradient term, $-\nabla_{\boldsymbol{\theta}} E_{\boldsymbol{\theta}}(s)$, is calculated by automatic differentiation in the deep network of the neural agent. Estimating $\nabla_{\boldsymbol{\theta}} \log Z_{\boldsymbol{\theta}}$ requires Markov chain Monte Carlo (MCMC) [84] sampling from the Boltzmann distribution within the inner loop of learning. Specifically, a sampler based on Stochastic Gradient Langevin Dynamics [23] (SGLD), a practical method of Langevin MCMC [85]) can effectively draw samples as:

$$\hat{s}^0 \sim \mathcal{U}, \quad \hat{s}^{i+1} = \hat{s}^i - \frac{\epsilon^2}{2} \nabla_{\hat{s}} E_{\boldsymbol{\theta}}(\hat{s}^i) + \xi, \quad \xi \sim \mathcal{N}, \tag{8}$$

where $\mathcal{U}$ denotes the uniform distribution, $i$ is the sampling iteration with step size $\epsilon$, $\xi$ is Gaussian noise, and $\hat{s}^i$ will converge to $P_{\boldsymbol{\theta}}(\hat{s})$ when $\epsilon \to 0$ and $i \to \infty$. While training on a new state, this MCMC chain will be reset. In practice, adopting standard MCMC sampling suffers from slow convergence and requires expensive computation. Recent persistent contrastive divergence approaches [76, 21] maintain a single MCMC chain, and use it to initialize a new chain for next sampling iteration [22].

Inspired by this, we introduce a Marginal State Memory $\mathcal{M}$ served as an online replay buffer [22], collecting historical information of the MCMC chain during training (Fig. 2). To initialize the inner loop of SGLD, the state $\hat{s}^0$ is sampled randomly from $\mathcal{U} \cup \mathcal{M}$. This helps continue to refine the previous samples, further accelerating the convergence. After several iterations $I$, $\hat{s}^I$ is stored in $\mathcal{M}$:

$$\hat{s}^0 \sim \mathcal{U} \cup \mathcal{M}, \quad \hat{s}^{i+1} = \hat{s}^i - \frac{\epsilon^2}{2} \nabla_{\hat{s}} E_{\boldsymbol{\theta}}(\hat{s}^i) + \xi, \quad \hat{s}^I \to \mathcal{M}, \tag{9}$$

where $I$ is fewer than the number of iterations required for MCMC convergence. The training details are illustrated in Algorithm 1.

### 3.3 Comparison with Other Imitation Learning Methods

In VLN, ENP aims to learn a policy $\pi^*$ from expert demonstrations ($\pi^e$) without access to the explicit reward, which is analogous to the most common approaches of IL, *e.g.*, BC and inverse reinforcement learning (IRL). For in-depth analysis, we explore the relation between ENP and other IL Methods, and provide a general understanding of them from a divergence viewpoint [28].

Standard BC is treated as a supervised learning problem (Eq. 1), and the policy is required to match the conditional distribution of $\pi^e$ under Kullback–Leibler (KL) divergence distance $D_{\mathrm{KL}}$ as:

$$\arg\min -\mathbb{E}[\log \pi^*(a|s)] = D_{\mathrm{KL}}(\pi^e(a|s) \| \pi^*(a|s)) - \mathbb{E}(\pi^e(a|s)), \tag{10}$$

where $\mathcal{H}(\pi^e) = \mathbb{E}(\pi^e(a|s))$ is a constant, *a.k.a.* the causal entropy [86]. $\mathcal{H}(\pi^e)$ is considered as the expected number of options at each state for $\pi^e(a|s)$ [28, 15].

---

**Algorithm 1** Energy-based Navigation Policy (ENP) Learning Algorithm

---

1: **Input**: The training dataset $\mathcal{D} = \{(x, \tau)\}$, Cross-Modal State Encoder $\mathcal{F}_{\boldsymbol{\theta}_s}$, Action Classifier $\mathcal{F}_{\boldsymbol{\theta}_a}$, Visual embedding $\boldsymbol{O}_t$, Instruction embedding $\boldsymbol{x}$, History embedding $\boldsymbol{H}_t$
2: **Initialize**: $\boldsymbol{\theta} = \boldsymbol{\theta}_s \cup \boldsymbol{\theta}_a$, and Marginal State Memory $\mathcal{M} \leftarrow \varnothing$ or $\mathcal{M}_{\text{ft}} \leftarrow \mathcal{M}_{\text{pre}}$
3: **for** training step $n = 1$ to $N$ **do**
4:     Aggregate $\mathcal{D}_n^+ = \{(x, \tau_n^+)\}$ of visited states by $\pi_n^*$ and actions from $\pi^{\text{e}}$. Sample $(x, \tau)$ from $\mathcal{D}_n^+ \cup \mathcal{D}$
5:     **for** navigation step $t = 1$ to $T$ **do**
6:         $\boldsymbol{s}_t = \mathcal{F}_{\boldsymbol{\theta}_s}(\boldsymbol{x}, \boldsymbol{O}_t, \boldsymbol{H}_t)$, $\boldsymbol{a}_t = \mathcal{F}_{\boldsymbol{\theta}_a}(\boldsymbol{s}_t)$, Sample $\hat{s}_t^0$ from $\mathcal{U} \cup \mathcal{M}$         $\triangleright$ Defined in §3.1.
7:         **for** SGLD iteration $i = 1$ to $I$ **do**
8:             $\hat{s}_t^{i+1} = \hat{s}_t^i - \frac{\epsilon^2}{2} \nabla_{\hat{s}} E_\theta(\hat{s}_t^i) + \xi$, $\xi \sim \mathcal{N}$, $\hat{s}_t^I \rightarrow \mathcal{M}$         $\triangleright$ Defined in Eq. (9).
9:         $\tilde{\mathcal{L}}_\pi \leftarrow -\log P_{\boldsymbol{\theta}}(a_t | \boldsymbol{s}_t)$, $\tilde{\mathcal{L}}_\mathcal{S} \leftarrow \left( E_{\boldsymbol{\theta}}(\boldsymbol{s}_t) - E_{\boldsymbol{\theta}}(\hat{s}_t^I) \right)$      $\triangleright$ Defined in Eq. (1) and (7).
10:     Backpropagate $\nabla_{\boldsymbol{\theta}} \mathcal{L}_\rho = \nabla_{\boldsymbol{\theta}} \mathcal{L}_\pi + \nabla_{\boldsymbol{\theta}} \mathcal{L}_\mathcal{S}$. Update $\mathcal{F}_{\boldsymbol{\theta}_s}$ and $\mathcal{F}_{\boldsymbol{\theta}_a}$
11: **Return**: Optimized Parameters $\boldsymbol{\theta} = \boldsymbol{\theta}_s \cup \boldsymbol{\theta}_a$ of $\mathcal{F}_{\boldsymbol{\theta}_s}$ and $\mathcal{F}_{\boldsymbol{\theta}_a}$

---

For ENP, the optimization objective in Eq. 3 can be written in a form of KL distance minimization:

$$\arg \min -\mathbb{E}_{(s,a) \sim \rho^{\text{e}}(s,a)}[\log \rho^*(s,a)] = D_{\text{KL}}\left( \rho^{\text{e}}(s,a) || \rho^*(s,a) \right) + \mathcal{H}(\rho^{\text{e}}(s,a)), \tag{11}$$

where $\mathcal{H}(\rho^{\text{e}}(s,a))$ is the entropy of the occupancy measure, which is a constant. IRL aims to infer the reward function of the expert, and learn a policy to optimize this reward. A representative work, AIRL [27], adopts an adversarial reward learning, which is equivalent to matching $\rho^{\text{e}}(s,a)$ [15] as:

$$\arg \min D_{\text{KL}}\left( \rho^*(s,a) || \rho^{\text{e}}(s,a) \right) = -\mathbb{E}_{\rho^*(s,a)}[\log \rho^{\text{e}}(s,a)] - \mathcal{H}(\rho^*(s,a)). \tag{12}$$

Hence, our objective is similar to the forward KL version of AIRL [28, 27] except for the entropy term. Both AIRL and ENP perform prioritized optimization for entire trajectories rather than the single-time step decisions in BC, so the compounding error can be mitigated [15, 27]. As $D_{\text{KL}}$ is a non-symmetric metric, the forward KL divergence results in distributions with a mode-covering behaviour, while using the reverse KL results in mode-seeking behaviour [18]. In addition, ENP directly models the joint distribution instead of recovering the reward function. We compare them in the experiment (§4.3) and find ENP is more applicable to VLN (Table 8).

### 3.4 Implementation Details

**Network Architecture.** ENP is a general framework that can be built upon existing VLN models. We investigate ENP on several representative architectures, including: i) EnvDrop [3] adopts a Seq2Seq model with a bidirectional LSTM-RNN and an attentive LSTM-RNN. ii) VLN↻BERT [4] is built upon multi-layer transformers with recurrent states. iii) DUET [6] is a dual-scale transformer-based model with a topological map for global action space, enabling long-term decision-making. iv) ETPNav [29] utilizes a transformer-based framework with an online topological map. For continuous environments, both VLN↻BERT [4] and ETPNav [29] employ a waypoint predictor [64]. $K_t$ dimension of the state representations $\boldsymbol{s}_t \in \mathbb{R}^{K_t \times D}$ is different due to the local or global action spaces of these methods. In §4, we follow the same settings of these models, replace their policy learning as ENP, and maintain a Marginal State Memory $\mathcal{M}$ of 100 buffer-size.

**Training.** For fairness, the hyper-parameters (*e.g.*, batch size, optimizers, maximal iterations, learning rates) of these models are kept the original setup. Back translation is employed to train EnvDrop [3]. The parameters of VLN↻BERT [4] is initialized from PREVALENT [51]. The pre-training and finetuning paradigm is used for DUET [6] and ETPNav [29]. For SGLD, $\hat{s}^0$ is sampled from $\mathcal{M}$ with a probability of 95% (Eq. 9). In EnvDrop [3] and VLN↻BERT [4], the number of SGLD iterations is set as $I = 15$. $I_{\text{pre}} = 20$ and $I_{\text{ft}} = 5$ of the pre-training and finetuning stages are used for DUET [6] and ETPNav [29]. Although the step size $\epsilon$ and the Gaussian noise $\xi$ in the original transition kernel of SGLD are related by $\text{Var}(\xi) = \epsilon$ [23], most studies [22, 79] choose to set $\epsilon$ and $\xi$ separately in practice. The step size $\epsilon = 1.5$ and $\xi \sim \mathcal{N}(0, 0.01)$ are set in ENP (§4.3).

**Inference.** During the testing phase, the agent uses the learned $\mathcal{F}_{\boldsymbol{\theta}_s}$ for cross-modal state encoding, and predicts the next step via $\mathcal{F}_{\boldsymbol{\theta}_a}$, until it chooses [STOP] action or reaches the maximum step limit.

**Reproducibility.** All experiments are conducted on a single NVIDIA 4090 GPU with 24GB memory in PyTorch. Testing is conducted on the same machine.

## 4 Experiment

We examine the efficacy of ENP for VLN in discrete environments (§4.1), and more challenging continuous environments (§4.2). Then we provide diagnostic analysis on core model design (§4.3).

### 4.1 VLN in Discrete Environments

**Datasets.** R2R [1] contains 7, 189 shortest-path trajectories captured from 90 real-world building-scale scenes [87]. It consists of 22K step-by-step navigation instructions. REVERIE [30] contains 21, 702 high-level instructions, which requires an agent to reach the correct location and localize a remote target object. All these datasets are devided into *train*, *val seen*, *val unseen*, and *test unseen* splits, which mainly focus on the generalization capability in unseen environments. Both R2R [1] and REVERIE [30] are built upon Matterport3D Simulator [1].

**Evaluation Metrics.** As in previous work [1, 30], Success Rate (SR), Trajectory Length (TL), Oracle Success Rate (OSR), Success rate weighted by Path Length (SPL), and Navigation Error (NE) are used for R2R. In addition, Remote Grounding Success rate (RGS) and Remote Grounding Success weighted by Path Length (RGSPL) are adopted for object grounding in REVERIE.

**Performance on R2R.** Table 1 demonstrates the numerical results of ENP with different frameworks on R2R. With Seq2Seq-style neural architectures, ENP provides **1**% SR and **1**% SPL gains over EnvDrop [3] on *val unseen*. Adopting ENP in transformer-based models leads to promising performance gains on *test unseen*, *e.g.*, **2**% on SR with VLN↺BERT [4], and **2**% on SR with DUET [6]. The performance is still lower than BEVBert [61] and BSG [62] due to the additional semantic maps. Moreover, we provide the average success rate across different path lengths in Appendix (Fig. 4).

Table 1: Quantitative comparison results on R2R [1] (§4.1).

| Models | R2R *val unseen* | | | | R2R *test unseen* | | | |
|---|---|---|---|---|---|---|---|---|
| | TL↓ | NE↓ | SR↑ | SPL↑ | TL↓ | NE↓ | SR↑ | SPL↑ |
| Seq2Seq [1] [CVPR2018] | 8.39 | 7.81 | 22 | – | 8.13 | 7.85 | 20 | 18 |
| SF [33] [NeurIPS2018] | – | 6.62 | 35 | – | 14.82 | 6.62 | 35 | 28 |
| AuxRN [48] [CVPR2020] | – | 5.28 | 55 | 50 | – | 5.15 | 55 | 51 |
| Active [88] [ECCV2020] | 20.60 | 4.36 | 58 | 40 | 21.60 | 4.33 | 60 | 41 |
| HAMT [5] [NeurIPS2021] | 11.46 | 2.29 | 66 | 61 | 12.27 | 3.93 | 65 | 60 |
| SSM [38] [CVPR2021] | 20.7 | 4.32 | 62 | 45 | 20.4 | 4.57 | 61 | 46 |
| HOP [89] [CVPR2022] | 12.27 | 3.80 | 64 | 57 | 12.68 | 3.83 | 64 | 59 |
| TD-STP [90] [MM2022] | – | 3.22 | 70 | 63 | – | 3.73 | 67 | 61 |
| LANA [91] [ICCV2023] | 12.00 | – | 68 | 62 | 12.60 | – | 65 | 60 |
| BSG [62] [ICCV2023] | 14.90 | 2.89 | 74 | 62 | 14.86 | 3.19 | 73 | 62 |
| BEVBert [61] [ICCV2023] | 14.55 | 2.81 | 75 | 64 | – | 3.13 | 73 | 62 |
| VER [92] [CVPR2024] | 14.83 | 2.80 | 76 | 65 | 15.23 | 2.74 | 76 | 66 |
| EnvDrop [3] [NAACL2019] | 10.70 | 5.22 | 52 | 48 | 11.66 | 5.23 | 51 | 47 |
| ENP–EnvDrop | 11.17 | 4.69 | 53 | 49 | 10.78 | 5.30 | **52** | **48** |
| VLN↺BERT [4] [CVPR2021] | 12.01 | 3.93 | 63 | 57 | 12.35 | 4.09 | 63 | 57 |
| ENP–VLN↺BERT | 12.17 | **3.81** | 65 | 57 | 13.12 | **3.91** | 65 | **59** |
| DUET [6] [CVPR2022] | 13.94 | 3.31 | 72 | 60 | 14.73 | 3.65 | 69 | 59 |
| ENP–DUET | 14.23 | **3.00** | **74** | 60 | 14.27 | 3.39 | 71 | **60** |

**Performance on REVERIE.** Table 2 presents the results of ENP with different transformer-based frameworks on REVERIE. ENP surpasses all the counterparts on both *val unseen* and *test unseen* by large margins, *e.g.*, **1.34**% on SR and **0.64**% on RGS over VLN↺BERT [4], **0.68**% on SR and **1.22**% on RGS over DUET [6], verifying the efficacy of ENP. Inspired by recent efforts [61], potential improvement on object grounding can be achieved by introducing CLIP features [93].

Table 2: Quantitative comparison results on REVERIE [30] (§4.1). '−': unavailable statistics.

| Models | REVERIE *val unseen* | | | | | | REVERIE *test unseen* | | | | | |
|---|---|---|---|---|---|---|---|---|---|---|---|---|
| | TL↓ | OSR↑ | SR↑ | SPL↑ | RGS↑ | RGSPL↑ | TL↓ | OSR↑ | SR↑ | SPL↑ | RGS↑ | RGSPL↑ |
| Seq2Seq [1] [CVPR2018] | 11.07 | 8.07 | 4.20 | 2.84 | 2.16 | 1.63 | 10.89 | 6.88 | 3.99 | 3.09 | 2.00 | 1.58 |
| RCM [8] [CVPR2019] | 11.98 | 14.23 | 9.29 | 6.97 | 4.89 | 3.89 | 10.60 | 11.68 | 7.84 | 6.67 | 3.67 | 3.14 |
| INP [30] [CVPR2020] | 45.28 | 28.20 | 14.40 | 7.19 | 7.84 | 4.67 | 39.05 | 30.63 | 19.88 | 11.61 | 11.28 | 6.08 |
| Airbert [94] [ICCV2021] | 18.71 | 34.51 | 27.89 | 21.88 | 18.23 | 14.18 | 17.91 | 34.20 | 30.28 | 23.61 | 16.83 | 13.28 |
| HAMT [5] [NeurIPS2021] | 14.08 | 36.84 | 32.95 | 30.20 | 18.92 | 17.28 | 13.62 | 33.41 | 30.40 | 26.67 | 14.88 | 13.08 |
| HOP [89] [CVPR2022] | 16.46 | 36.24 | 31.78 | 26.11 | 18.85 | 15.73 | 16.38 | 33.06 | 30.17 | 24.34 | 17.69 | 14.34 |
| TD-STP [90] [MM2022] | – | 39.48 | 34.88 | 27.32 | 21.16 | 16.56 | – | 40.26 | 35.89 | 27.51 | 19.88 | 15.40 |
| LANA [91] [ICCV2023] | 23.18 | 52.97 | 48.31 | 33.86 | 32.86 | 22.77 | 18.83 | 57.20 | 51.72 | 36.45 | 32.95 | 22.85 |
| BSG [62] [ICCV2023] | 24.71 | 58.05 | 52.12 | 35.59 | 35.36 | 24.24 | 22.90 | 62.83 | 56.45 | 38.70 | 33.15 | 22.34 |
| BEVBert [61] [ICCV2023] | – | 56.40 | 51.78 | 36.37 | 34.71 | 24.44 | – | 57.26 | 52.81 | 36.41 | 32.06 | 22.09 |
| VER [92] [CVPR2024] | 23.03 | 61.09 | 55.98 | 39.66 | 33.71 | 23.70 | 24.74 | 62.22 | 56.82 | 38.76 | 33.88 | 23.19 |
| VLN↺BERT [4] [CVPR2021] | 16.78 | 35.02 | 30.67 | 24.90 | 18.77 | 15.27 | 15.86 | 32.91 | 29.61 | 23.99 | 16.50 | 13.51 |
| ENP–VLN↺BERT | 16.90 | 35.80 | 31.27 | 25.22 | 18.78 | 15.80 | 15.90 | **34.00** | 30.95 | 24.46 | 17.14 | 14.12 |
| DUET [6] [CVPR2022] | 22.11 | 51.07 | 46.98 | 33.73 | 32.15 | 23.03 | 21.30 | 56.91 | 52.51 | 36.06 | 31.88 | 22.06 |
| ENP–DUET | 25.76 | **54.70** | 48.90 | 33.78 | 34.74 | 23.39 | 22.70 | **59.38** | 53.19 | 36.26 | 33.10 | 22.14 |

**Qualitative Results.** In Fig. 3 (a), we illustrate the qualitative comparisons of ENP against DUET [6] on R2R *val unseen*. Based on ENP, our agent yields more accurate predictions than DUET. It verifies that ENP leads to better decision-making when facing challenging scenes. In Fig. 3 (b), our agent may fail due to the serious occlusion, and introducing map prediction [95] may alleviate this problem.

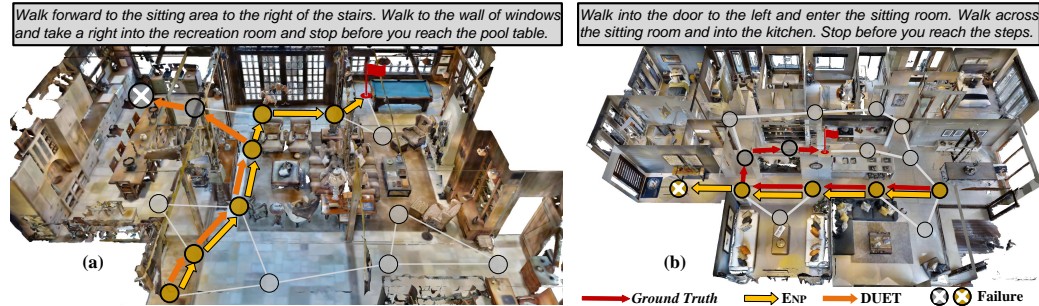

Figure 3: Qualitative results on R2R [1] (§4). (a) DUET [6] arrives in the wrong room instead of 'recreation room' since the scene contains multiple rooms. Our agent reaches the goal successfully, demonstrating better decision-making ability. (b) Failure case: Due to partial observability and occlusion of the environment, it is hard to find 'kitchen' at some positions. Thus our agent goes the wrong way and ends in failure (§4.1).

## 4.2 VLN in Continuous Environments

**Datasets.** R2R-CE [31] and RxR-CE [32] are more practical yet challenging, where the agent can navigate to any unobstructed point instead of teleporting between fixed nodes of a pre-defined navigation graph in R2R. R2R-CE comprises a total of 5, 611 shortest-path trajectories with significantly longer time horizons than R2R. RxR-CE presents more instructions with longer annotated paths, including multilingual descriptions. In addition, the agents in RxR-CE are forbidden to slide along obstacles. Both of are R2R-CE and RxR-CE driven by Habitat Simulator [96].

**Evaluation Metrics.** The metrics of R2R-CE [31] are similar to R2R [1]. For RxR-CE [32], Normalize Dynamic Time Wrapping (NDTW) and NDTW penalized by SR (SDTW) are further used to evaluate the fidelity between the predicted and annotated paths.

**Performance on R2R-CE.** As shown in Table 3, ENP provides a considerable performance gain against VLN↻BERT [4], *e.g.*, **1**% SR and **1**% SPL on *val unseen*. Concretely, it outperforms the top-leading ETPNav [29] by **1**% and **1**% in terms of OSR and SR on *test unseen*, respectively. These quantitative results substantiate our motivation of empowering VLN agents with ENP.

Table 3: Quantitative comparison results on R2R-CE [31] (§4.2). '−': unavailable statistics.

| Models | R2R-CE | | | | | | | | | | | | | | |
|---|---|---|---|---|---|---|---|---|---|---|---|---|---|---|---|
| | *val seen* | | | | | *val unseen* | | | | | *test unseen* | | | | |
| | TL↓ | NE↓ | OSR↑ | SR↑ | SPL↑ | TL↓ | NE↓ | OSR↑ | SR↑ | SPL↑ | TL↓ | NE↓ | OSR↑ | SR↑ | SPL↑ |
| VLN-CE [31] [ECCV2020] | 9.26 | 7.12 | 46 | 37 | 35 | 8.64 | 7.37 | 40 | 32 | 30 | 8.85 | 7.91 | 36 | 28 | 25 |
| CMTP [37] [CVPR2021] | − | 7.10 | 56 | 36 | 31 | − | 7.90 | 38 | 26 | 23 | − | − | − | − | − |
| HPN [97] [ICCV2021] | 8.54 | 5.48 | 53 | 46 | 43 | 7.62 | 6.31 | 40 | 36 | 34 | 8.02 | 6.65 | 37 | 32 | 30 |
| SASRA [98] [ICPR2022] | 8.89 | 7.71 | − | 36 | 34 | 7.89 | 8.32 | − | 24 | 22 | − | − | − | − | − |
| CM2 [99] [CVPR2022] | 12.05 | 6.10 | 51 | 43 | 35 | 11.54 | 7.02 | 42 | 34 | 28 | 13.90 | 7.70 | 39 | 31 | 24 |
| WSMGMap [60] [NeurIPS2022] | 10.12 | 5.65 | 52 | 47 | 43 | 10.00 | 6.28 | 48 | 39 | 34 | 12.30 | 7.11 | 45 | 35 | 28 |
| Sim2Sim [100] [ECCV2022] | 11.18 | 4.67 | 61 | 52 | 44 | 10.69 | 6.07 | 52 | 43 | 36 | 11.43 | 6.17 | 52 | 44 | 37 |
| BEVBert [61] [ICCV2023] | − | − | − | − | − | − | 4.57 | 67 | 59 | 50 | − | 4.70 | 67 | 59 | 50 |
| DREAM [71] [ICCV2023] | 11.60 | 4.09 | 66 | 59 | 48 | 11.30 | 5.53 | 59 | 49 | 44 | 11.80 | 5.48 | 57 | 49 | 44 |
| HNR [101] [CVPR2024] | 11.79 | 3.67 | 76 | 69 | 61 | 12.64 | 4.42 | 67 | 61 | 51 | 13.03 | 4.81 | 67 | 58 | 50 |
| VLN↻BERT [4] [CVPR2021] | 12.50 | 5.02 | 59 | 50 | 44 | 12.23 | 5.74 | 53 | 44 | 39 | 13.51 | 5.89 | 51 | 42 | 36 |
| ENP–VLN↻BERT | **12.08** | **4.89** | 60 | 51 | 44 | 12.37 | 5.81 | 53 | **45** | **40** | **13.21** | **5.68** | 52 | 44 | 36 |
| ETPNav [29] [TPAMI2024] | 11.78 | 3.95 | 72 | 66 | 59 | 11.99 | 4.71 | 65 | 57 | 49 | 12.87 | 5.12 | 63 | 55 | 48 |
| ENP–ETPNav | 11.82 | **3.90** | 73 | 68 | 59 | **11.45** | **4.69** | 65 | **58** | **50** | **12.71** | **5.08** | 64 | 56 | 48 |

**Performance on RxR-CE.** Table 4 reports the results with Marky-mT5 instructions [56] on RxR-CE. ENP achieves competitive results on longer trajectory navigation with multilingual instructions, *e.g.*, **55.27**% *vs.* 54.79% SR and **62.97**% *vs.* 61.90% NDTW for ETPNav [29] on *val unseen*. We attribute this to the global expert policy matching, which promotes the path fidelity. This indicates ENP enables the agent to make long-term plans, resulting in better NDTW and SDTW.

Table 4: Quantitative comparison results on RxR-CE [32] (§4.2). '−': unavailable statistics.

| Models | RxR-CE *val seen* | | | | | RxR-CE *val unseen* | | | | |
|---|---|---|---|---|---|---|---|---|---|---|
| | NE↓ | SR↑ | SPL↑ | NDTW↑ | SDTW↑ | NE↓ | SR↑ | SPL↑ | NDTW↑ | SDTW↑ |
| CWP-CMA [64] [CVPR2022] | − | − | − | − | − | 8.76 | 26.59 | 22.16 | 47.05 | − |
| VLN↻BERT [4] [CVPR2021] | − | − | − | − | − | 8.98 | 27.08 | 22.65 | 46.71 | − |
| Reborn [29] [CVPR2022] | 5.69 | 52.43 | 45.46 | 66.27 | 44.47 | 5.98 | 48.60 | 42.05 | 63.35 | 41.82 |
| HNR [101] [CVPR2024] | 4.85 | 63.72 | 53.17 | 68.81 | 52.78 | 5.51 | 56.39 | 46.73 | 63.56 | 47.24 |
| ETPNav [29] [TPAMI2024] | 5.03 | 61.46 | 50.83 | 66.41 | 51.28 | 5.64 | 54.79 | 44.89 | 61.90 | 45.33 |
| ENP–ETPNav | 5.10 | **62.01** | **51.18** | **67.22** | **51.90** | **5.51** | **55.27** | **45.11** | **62.97** | **45.83** |

### 4.3 Diagnostic Experiment

**Joint Distribution Matching.** We first investigate the joint distribution matching loss $\mathcal{L}_\rho$ (Eq. 3) in ENP, which is factored as the discriminative action loss $\mathcal{L}_\pi$ (Eq. 1) and the marginal state distribution matching loss $\mathcal{L}_\mathcal{S}$ (Eq. 7). Both of $\mathcal{L}_\pi$ and $\mathcal{L}_\mathcal{S}$ are online optimized iteratively. A clear performance

Table 5: Ablation study of objective loss (§4.3).

| Models | Objective | R2R val | | R2R-CE val | |
|---|---|---|---|---|---|
| | | SR↑ | SPL↑ | SR↑ | SPL↑ |
| ENP–VLN◯BERT [4] | $\mathcal{L}_\pi$ | 61 | 55 | 44 | 39 |
| | $\mathcal{L}_\pi + \mathcal{L}_\mathcal{S}$ | **65** | **57** | **45** | **40** |
| ENP–DUET&ETPNav [6, 29] | $\mathcal{L}_\pi$ | 72 | 60 | 57 | 49 |
| | $\mathcal{L}_\pi + \mathcal{L}_\mathcal{S}$ | **74** | 60 | **58** | **50** |

drop is observed in Table 5, *e.g.*, SR from **74**% to 72% for DUET [6] on R2R, as only optimizing the conditional action distribution with $\mathcal{L}_\pi$ will degenerate into BC (or DAGGER) with cross-entropy loss. This reveals the appealing effectiveness of the joint distribution matching strategy in ENP.

**Step Size and Guassian Noise in SGLD.** In ENP, SGLD [23] is used to draw samples from Boltzmann distribution (Eq. 9) and further optimize $\mathcal{L}_\mathcal{S}$. Following recent EBM training, we relax the restriction on the relation between the step size $\epsilon$ and Gaussian noise $\xi$, *i.e.*, $\text{Var}(\xi) = \epsilon$.

Table 6: Ablation study of step size and noise (§4.3).

| Models | #SGLD | R2R val | | R2R-CE val | |
|---|---|---|---|---|---|
| | | SR↑ | SPL↑ | SR↑ | SPL↑ |
| ENP–VLN◯BERT [4] | $\epsilon = 1$, $\text{Var}(\xi) = 0.01$ | 64 | 57 | 45 | 39 |
| | $\epsilon = 1.5$, $\text{Var}(\xi) = 0.01$ | **65** | **57** | **45** | **40** |
| | $\epsilon = 2$, $\text{Var}(\xi) = 0.01$ | 64 | 56 | 44 | 40 |
| | $\epsilon = 1.5$, $\text{Var}(\xi) = 0.1$ | 63 | 55 | 41 | 38 |

Table 6 shows the results with different step sizes (when $I = 15$), and changing $\epsilon$ in a small range has little effects on the performance. In addition, different noise variances $\text{Var}(\xi)$ are also explored. Our ENP is insensitive to parameter changes. We find that $\epsilon = 1.5$ and $\xi \sim \mathcal{N}(0, 0.01)$ for the transition kernel of SGLD works well across a variety of datasets and frameworks in VLN.

**Number of SGLD Loop per Training Step.** For the SGLD sampler, we study the influence of the number of inner loop iterations per training step. In Table 7, we can find that $I = 15$ iterations is enough to sample the recurrent states for En-vDrop [3] and VLN◯BERT [4]. Slightly more iterations are required for the models [6, 29] with

Table 7: Ablation study of SGLD iterations (§4.3).

| Models | #Loop | R2R val | | R2R-CE val | |
|---|---|---|---|---|---|
| | | SR↑ | SPL↑ | SR↑ | SPL↑ |
| ENP–VLN◯BERT [4] | $I = 5$ | 64 | 56 | 44 | 39 |
| | $I = 15$ | **65** | **57** | **45** | **40** |
| | $I = 20$ | 65 | 57 | 45 | 40 |
| ENP–DUET [6] &ENP–ETPNav [29] | $I_{\text{pre}} = 10$, $I_{\text{ft}} = 5$ | 74 | 58 | 56 | 49 |
| | $I_{\text{pre}} = 20$, $I_{\text{ft}} = 5$ | **74** | **60** | **58** | **50** |
| | $I_{\text{pre}} = 20$, $I_{\text{ft}} = 10$ | 73 | 60 | 58 | 49 |

additional topological graph memory due to the global action space. This can be mitigated by pre-storing the samples in the Marginal State Memory $\mathcal{M}$ during pre-training. As these frameworks [6, 29] employ pre-training and finetuning paradigm for VLN, we design a different number of iterations, *i.e.*, $I_{\text{pre}} = 20$ and $I_{\text{ft}} = 5$. It is worth noting that the state memory $\mathcal{M}_{\text{pre}}$ can acquire the samples while performing single-step action prediction in the VLN pre-training. Thus, the SGLD iterations in finetuning based on $\mathcal{M}_{\text{ft}} \leftarrow \mathcal{M}_{\text{pre}}$ can be further reduced.

**ENP *vs.* AIRL.** In Table 8, we compare ENP against AIRL [27], which is a representative work of adversarial imitation learning. The objective function for ENP is equivalent to the forward KL divergence minimization, where

Table 8: ENP *vs.* AIRL (§4.3).

| Models | Policy | R2R val | | R2R-CE val | |
|---|---|---|---|---|---|
| | | SR↑ | SPL↑ | SR↑ | SPL↑ |
| VLN◯BERT [4] | AIRL[27] | 63 | 57 | 43 | 39 |
| | ENP | **65** | 57 | **45** | **40** |

AIRL [27] makes use of the backward KL divergence. By modifying the original policy learning in VLN◯BERT [4] (a mixture of RL and IL) as an adversarial reward learning formulation, AIRL achieves the comparable performance. ENP yields better scores on R2R-CE with **45**% SR and **40**% SPL. Meanwhile, ENP avoids the potential influence from the adversarial training.

**Runtime Analysis.** The additional computational complexity of ENP is from the iterations of SGLD inner loop. Towards this, we make some specific designs, such as Marginal State Memory $\mathcal{M}$ and pre-storing in pre-training. During inference, there is no additional computational consumption in executing the strategy compared to existing VLN agents.

## 5 Conclusion

In this paper, we propose ENP, an Energy-based Navigation Policy learning framework. By explicitly modeling the joint state-action distribution using an energy-based model, ENP shows promise to solve the intrinsic limitations of supervised behavioural cloning. Through maximizing the likelihood of the action prediction and modeling the dynamics of the navigation states jointly, ENP learns to minimize the divergence distance from the expert policy in a collaborative manner. Experimental results on gold-standard VLN datasets in both discrete (R2R and REVERIE) and continuous environments (R2R-CE and RxR-CE) demonstrate the superiority against existing methods. In the future, we will explore more tasks and frameworks with the energy-based policy.

**Acknowledgment.** This work was supported by the National Science and Technology Major Project (No. 2023ZD0121300), the National Natural Science Foundation of China (No. 62372405), the Fundamental Research Funds for the Central Universities 226-2024-00058, the National Key Laboratory of Human-Machine Hybrid Augmented Intelligence, Xi'an Jiaotong University (No. HMHAI-202403), CIPSC-SMP-Zhipu Large Model Cross-Disciplinary Fund, and the Earth System Big Data Platform of the School of Earth Sciences, Zhejiang University.

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

# A Appendix

## A.1 Model Details

**EnvDrop** [3] adopts an encoder-decoder model. The encoder is a bidirectional LSTM-RNN with an embedding layer. The decoder is an attentive LSTM-RNN. EnvDrop uses the 2048-dimensional image features from ResNet-152 pretrained on ImageNet. The model is first trained with supervised learning via the mixture of imitation and reinforcement learning. Then, it is finetuned by back translation with environmental dropout. The word embedding size is 256 and the dimension of the action embedding is 64. The size of the LSTM units is set to 512 while 256 is set for the bidirectional LSTM. RMSprop [102] is employed to optimize the loss with a learning rate $1\times10^{-4}$ and the batch size is set as 64.

**VLN↻BERT** [4] is built upon LXMERT-like transformer [103]. It is trained directly on the mixture of the original training data and the augmented data from PREVALENT [51] for R2R [1]. AdamW optimizer [104] is used and the batch size is set to 16. For REVERIE [30], the image features are extracted by ResNet-152 pre-trained on Places365 [48], and the object features are encoded by Faster-RCNN pre-trained on Visual Genome [105].The batch size is set as 8.

**VLN↻BERT for CE** [64] proposes a waypoint predictor to generate a set of candidate waypoints during navigation. The training objective is the cross-entropy loss between the ground-truth and action predictions. It is minimized by AdamW optimizer [104] with the learning rate $1\times10^{-5}$ during training. The decay ratio is set as 0.50 for R2R-CE [31] and 0.75 for RxR-CE [32].

**DUET** [6] uses a dual-scale graph transformer initialized from LXMERT [103]. The hidden layer size is 768. The image features are extracted by ViT-B/16 [106] pretrained on ImageNet [107]. On REVERIE [30], DUET is first pre-trained with the batch size of 32. Some auxiliary tasks are employed, such as single-step action prediction, object grounding, masked language modeling, and masked region classification [5]. Then it is finetuned with the batch size of 8.

**ETPNav** [29] contains a topological mapping module and a waypoint prediction module [64]. It is pre-trained using proxy tasks following existing transformer-based VLN models [4, 6] with a batch size of 64 and a learning rate of $5\times10^{-5}$. During finetuning, the AdamW optimizer [104] is utilized. The batch size is set as 16 with a learning rate of $1\times10^{-5}$. The decay ratio is 0.75.

## A.2 Comparison Results across Different Path Lengths

In Fig. 4, ENP demonstrates notable performance improvements over BC [6] across both *seen* and *unseen* splits of R2R [1]. When the length of ground-truth paths exceeds 16 meters, our ENP outperforms BC by a significant margin in both the *val seen* and *val unseen* splits. In addition, the performance gains are relatively more significant in the *unseen* split compared to the *seen* split. For example, ENP achieves approximately **5**% improvement of SR on the *val seen* and **6**% improvement on the *val unseen* for navigation paths longer than 16 meters.

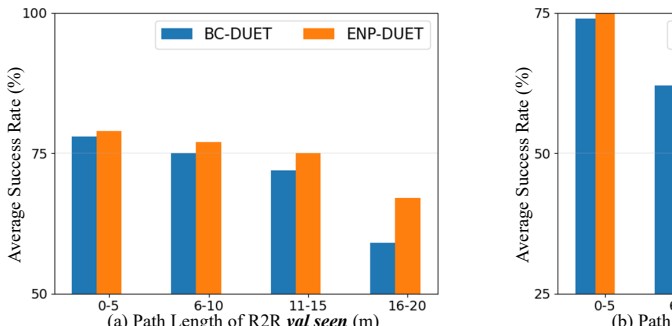
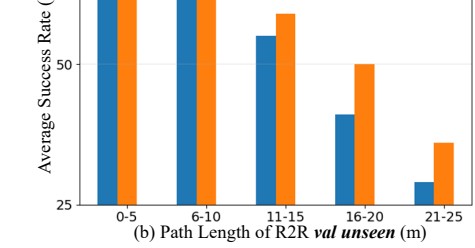

Figure 4: The average success rate of ENP and BC [6] across different trajectory lengths.

## A.3 Discussion

**Terms of use, Privacy, and License.** R2R [1], REVERIE [30], R2R-CE [31], and RxR-CE [32] datasets utilized in our work contain a large number of indoor photos from Matterport3D [87],

which are freely available for non-commercial research purposes. No identifiable individuals are present in any of the photographs. All experiments are conducted on the Matterport3D and Habitat Simulators [1, 96], which are released under the MIT license.

**Limitations.** Existing studies in VLN mainly focus on improving the success rate of navigation, with limited attention given to collision avoidance and navigation safety. In addition, current agents are typically designed for static environments within simulators. To mitigate the risk of collisions, it is essential for robots to operate under specific security constraints. Therefore, further advancements are necessary to ensure safe deployment in real-world dynamic environments.

**Future Direction.** i) We explore ENP on several representative VLN architectures [3, 4, 6, 29], and the generalization of ENP to other architectures [63, 61] and more tasks [108] is not clear. We will verify it in future work. ii) Efficiently sampling from un-normalized distributions for high-dimensional data is a fundamental challenge in machine learning. Utilizing MCMC-free approaches [75, 109] and neural implicit samplers [110] may offer valuable insights to enhance our approach. One limitation of ENP is that the optimization through the SGLD inner loop at each training iteration (Algorithm 1), which would reduce the training efficiency (about 9% delay for 5 SGLD iterations). In practice, ENP balances performance and training time with a limited number of iterations. iii) In the future, agents should be able to actively acquire the skills by watching videos similar to humans [52], thereby reducing high training data (expert demonstrations) requirements.

**Broader Impacts.** We introduce an energy-based policy learning for VLN. Equipped with ENP, existing agents are able to perform comprehensive decision-making, showcasing promising improvements across various VLN benchmarks. In real-world deployment, the agent holds potential to significantly benefit society by providing assistance to individuals with specific needs, *e.g.*, blind and disabled, enabling them to accomplish daily tasks more independently. In addition, we encourage more efforts devoted to policy learning for future research in the community.

