# OpenReview forum: "Vision-Language Navigation with Energy-Based Policy"
_NeurIPS.cc/2024/Conference — NeurIPS 2024 poster_

### Official Review · Reviewer_dJ3y · 2024-07-02

**Soundness:** 2
**Presentation:** 3
**Contribution:** 2
**Rating:** 5
**Confidence:** 3

**Summary:**

To attempt to solve the catastrophic failure issues that arise from behavioral cloning, the authors introduce Energy-based Navigation Policy (Enp) which uses an energy-based model to learn the expert policy from demonstrations. The authors claim that by doing so Enp prioritizes learning the full trajectory instead of single-time step decisions in BC.

**Strengths:**

**S1.** The issue of catastrophic failure from BC is an important one that which is likely applicable to the setting of VLN — this justifies the importance of the problem.

**S2.** The method is simple and described appropriately, both of which are strengths of the work.

**Weaknesses:**

**W1.** *Overall improvements are low.* The performance improvements provided by ENP are small compared to the BC baselines, often around $1-2\%$ in terms of SR and similar in SPL. This is particularly poignant when compared to the improvements provided by semantic grid maps. By itself, this would not necessarily be a major issue, if it was clear where ENP was helping (W2).

**W2.** *It is unclear when ENP is better than BC, and if the intuitive justification of catastrophic failure applies.* A more in-depth analysis of the success/failure cases of using ENP vs BC is warranted. Metrics like SR suggest that ENP can be helpful in avoiding the failure that BC can induce, but it is not entirely clear this is the case — when is ENP helping? Given the justification of the catastrophic failure of BC, you would expect this would be in longer trajectories, but the example in Figure 3a does not justify that given its short trajectory length. Could you create a benchmark specifically to test this? Including this analysis could boost the case for the motivation behind ENP significantly.

**W3.** *ENP is likely more expensive to train than BC.* The authors should include a compute comparison for the training of ENP vs BC. While the inference runtime is the same, training time for ENP will likely be significantly influenced by $I$.

The combination of W1 and W2 severely hinder the practical motivation of the method.

Minor comments:

- It’s not really required to include all baselines in the main results tables. E.g., Seq2Seq, SF and RCM are pretty outdated at this point and always beaten by other methods — it’s better to focus the comparison with the most recent methods and leave the full tables to the appendix.

**Questions:**

**Q1.** Why are the DUET or, e.g., BEVBert/GridMM, paths not shown in Figure 3b too? If multiple agents face the same issue, that would show this is the problem of observability.

**Q2.** What are the $\pm$ values in Table 3, is it the standard deviation? If so, why is it not reported for the other methods?

**Q3.** Do the authors have any intuition on why increasing $I_{\text{pre}}$ and $I_{\text{ft}}$ could result in a lower SR/SPL (Table 7)?

**Limitations:**

The authors do not discuss the limitations of their method. The limitations paragraph in Appendix A.2. does not reflect the particular ENP method they propose and is instead generic to the problem. Training time is one potential limitation that should be studied and discussed in further detail.

---

> ### Author Rebuttal · Authors · 2024-08-07
>
> **Q1: The improvement of ENP is lower than that of semantic grid maps.**\
> **A1:** **First**, respectfully, the improvement of 1-2% on both SR and SPL is **non-trival** in VLN. This point was acknowledged by the other reviewers, such as "work pretty well" (Reviewer UeCa) and "achieving obvious improvements" (Reviewer Ngjf). **Second**, although we agree that semantic maps [54-57] are powerful tools, ENP improves the performance from different perspectives. ENP learns an unbiased expert policy by exploiting information from the training data (Eq. 3), while semantic maps introduce additional scene information, *e.g.*, [55] encodes spatial priors by grid maps. **Third**, the below table shows ENP requires significantly less GPU memory than [55] (DUET-based [6] framework).
>
> |Models|GPU Memory on NVIDIA RTX 4090|
> |:-:|:-:|
> |DUET [6]|14G|
> |BEVBert [55]|20G|
> |ENP-DUET|**15G**|
>
> Therefore, it is **unfair** to compare ENP directly against the semantic maps. More importantly, ENP is a flexible solution and can be seamlessly integrated into existing methods without extra inference time. The following table indicates that ENP is **complementary** to [55] and further enhance the performance.
>
> |Models|R2R *val unseen* SR|R2R *val unseen* SPL|REVERIE *val unseen* SR|REVERIE *val unseen* RGS|
> |:-:|:-:|:-:|:-:|:-:|
> |BEVBert|75|64|51.78|34.71|
> |ENP-BEVBert|**76**|**65**|**52.71**|**34.80**|
>
> **Q2: *"... when ENP is better than BC ..."***\
> **A2:** **First**, in Tables 1-4, ENP can achieve consistent performance gains across different datasets. **Second**, we explore ENP and BC across different trajectory lengths in Fig. S1 of the "global" response PDF. In Fig. S1 (a) and (b), ENP exceeds BC by **6**% and **9**% on R2R *val seen* and *val unseen* (16-20m). **Third**, we further validate the effectiveness of ENP on R4R [ref-3] with the longer paths in the following table (also in Fig. S1 (c) and (d)). ENP surpasses BC by **6.40**% and **9.03**% on R4R *val seen* and *val unseen* (>20m), and the gains in *unseen* environments are relatively more significant.
>
> |Models on R4R *val*|*seen* (<20m)|*seen* (>20m)|*unseen* (<20m)|*unseen* (>20m)|
> |:-:|:-:|:-:|:-:|:-:|
> |BC-DUET|70.88|57.25|52.52|43.71|
> |ENP-DUET|72.67|63.65|54.91|52.74|
> |$\Delta$|+1.79|+**6.40**|+2.39|+**9.03**|
>
> We will add these figures and tables to the Appendix.
>
> [ref-3] Stay on the Path: Instruction Fidelity in Vision-and-Language Navigation. ACL 2019.
>
> **Q3: A benchmark with longer trajectories.**\
> **A3:** We clarify that we have conducted experiments on R2R-CE [32] and RxR-CE [33] in Tables 3 and 4 (L283-292), which involve **significantly longer action trajectories** (56 vs. 5 steps in R2R). **In addition**, per your request, we report the results on R4R [ref-3] with longer paths. CLS [ref-3] measures how closely the trajectory aligns with the ground truth. ENP achieves **3.52**% and **3.11**% gains on SR and CLS of *val unseen*.
>
> |Models on R4R *val unseen*|SR|CLS|
> |:-:|:-:|:-:|
> |DUET|50.18|40.54|
> |ENP-DUET|**53.70**|**43.65**|
>
> **Q4: A detailed analysis of success/failure cases.**\
> **A4:** Thanks for your feedback. We include more success/failure cases in the PDF material of "global" response (Fig. S2). In Fig. S2 (a), our agent with ENP successfully reaches the final location, whereas DUET fails due to the accumulation of errors over long paths. In (b), both our agent and DUET struggle to locate the target due to the lack of a specific landmark recognition module like [50,51]. This figure will be added to Fig. 3.
>
> **Q5: Training time for ENP.**\
> **A5:** We report the training time for ENP with different SGLD iterations ***I*** on a NVIDIA RTX 4090. In practice, we find that ***$I_{ft}=5$*** SGLD loops per training iteration is good enough for model convergence (Algorithm 1), introducing a minor computational overhead, approximately 4 minutes delay for 1K training iterations.
>
> |Models|minutes/1K training iterations|SGLD *$I_{ft}$*|
> |:-:|:-:|:-:|
> |DUET|40|--|
> |ENP|42|3|
> |ENP|44|5|
> |ENP|54|10|
>
> **Q6: The comparison with recent methods.**\
> **A6:** Sure, we will reorganize Tables 1-3 and highlight the comparison with the most recent methods. The complete tables will be included in the Appendix.
>
> **Q7: *"... the DUET paths not shown in Figure 3b ..."***\
> **A7:** To clearly present the comparison, we only include two navigation trajectories in each figure for better understanding (Fig. 3). We will include the comparison results of these methods in the Appendix and provide further analysis.
>
> **Q8: The standard deviation ($\pm$) in tables.**\
> **A8:** Our apologies. "$\pm$" in Tables 1-4 denotes the standard deviation after repeating the experiments with different random seeds. Unfortunately, the corresponding standard deviations for other methods were not provided in their original papers.
>
> **Q9: *"... increasing $I_{pre}$ and $I_{ft}$ results in a lower SR/SPL (Table 7)"***\
> **A9:** Since there is a domain gap between *seen* and *unseen* environments [3,9], increasing SGLD iterations ($I_{ft}$) may lead to overfitting on *val seen*, resulting a lower SR/SPL on *val unseen*. Additional results on *val seen* for Table 7 are provided in the following.
>
> |Models|#Loop|R2R *val seen* SR|R2R *seen* SPL|R2R *val unseen* SR|R2R *unseen* SPL|R2R-CE *val seen* SR|R2R-CE *seen* SPL|R2R-CE *val unseen* SR|R2R-CE *unseen* SPL|
> |:-:|:-:|:-:|:-:|:-:|:-:|:-:|:-:|:-:|:-:|
> |DUET [6,30]|--|77|68|72|60|66|59|57|49|
> |ENP|$I_{pre}$=20, $I_{ft}$=5|78|68|**74**|**60**|68|59|**58**|**50**|
> |ENP|$I_{pre}$=20, $I_{ft}$=10|**81**|**70**|73|60|**69**|**60**|58|49|
>
>
> **Q10: More discussion about the limitations.**\
> **A10:** One limitation of ENP is that the optimization through the SGLD inner loop at each training iteration (Algorithm 1). It would reduce the training efficiency (about 9% delay for 5 SGLD iterations). In practice, ENP balances performance and training time with a limited number of SGLD iterations. We will provide more analysis in the Discussion.

---

> > ### Comment · Reviewer_dJ3y · 2024-08-12
> >
> > I thank the authors for their clear and detailed rebuttal. I believe the extra results provided make the paper much stronger than the original submission, particularly the ones on the success rate as a function of path length for ENP compared to BC and the training time table + discussion. It is key that these are incorporated in the paper to provide a full picture of the method's advantages and drawbacks. I will raise my original score.

---

> > > ### Author Response · Authors · 2024-08-12
> > >
> > > Dear Reviewer dJ3y,
> > >
> > > We greatly appreciate your valuable time and constructive feedback. We will incorporate your suggestions into the revised version of the paper. If you have any further questions or suggestions, please feel free to share them.
> > >
> > > Thanks again,\
> > > Authors

---

### Official Review · Reviewer_Ngjf · 2024-07-05

**Soundness:** 3
**Presentation:** 4
**Contribution:** 3
**Rating:** 7
**Confidence:** 2

**Summary:**

This paper studies vision-and-language navigation (VLN) by addressing the problems of error accumulation in the Markov decision process and the difficulties in designing an optimal reward function, respectively, in the commonly applied behavioral cloning (BC) and reinforcement learning (RL) approaches. It proposes an energy-based policy (ENP) for training the agent by estimating the unnormalized probability density of the expert’s occupancy measure, optimizing the entire trajectories rather than the single-time step decisions in BC. Meanwhile, to facilitate the MCMC sampling in ENP, a Marginal State Memory is introduced for the agent as an online reply buffer. The proposed method has been evaluated on multiple popular VLN agents and several benchmarking datasets and environments, together with detailed analyses of the ENP and design choices.

**Strengths:**

1.	The energy-based navigation policy proposed in this paper introduces a new training perspective for VLN and potentially for broader visual navigation problems. The idea is very well-motivated as it addresses the limitations of commonly applied BC and RL approaches. As a researcher in visual navigation, I am happy to see the exploration of such a fundamentally different training approach and the fact that the method works well across models and settings.
2.	Arguments and design choices made in this paper are well-justified by analyses and experiments (e.g., Section 3.3 and Section 4.3).
3.	The proposed idea has been evaluated across various popular VLN baselines (e.g., EnvDrop, VLN-BERT, and DUET), different datasets (R2R, REVERIE, and RxR), and different environments (e.g., discrete and continuous), achieving obvious improvements and showing the effectiveness of the method.
4.	Overall, the paper is nicely written. Contents, tables, and figures are clearly presented.

**Weaknesses:**

1.	The proposed ENP has only been evaluated on a limited scale of VLN datasets and models. It is unclear if the method will still benefit larger models, or the case where more sufficient training data is available to learn a very generalizable policy through IL/BC (e.g., 4.9M ScaleVLN data [49] vs. the original 14K R2R data). I am concerned that IL/BC is much more friendly and stable than ENP to scale with data.
2.	There is no comprehensive successful/failure case analysis of ENP versus existing training methods. It is unclear exactly in what scenarios an agent trained with ENP shows an advantage.

I need to mention that I have very little knowledge about energy-based models, so I might have missed some key details in mathematics. I will rely more (and probably adjust my review based) on the comments from the other reviewers on the theoretical part of this paper.

**Questions:**

No other questions; I wish the authors could kindly respond to my concerns in Weaknesses.

**Limitations:**

The authors have discussed some limitations of the proposed method and its potential societal impact. As for suggestions, mainly the two points in Weaknesses – test on large-scale training data and run thorough result analysis.

---

> ### Author Rebuttal · Authors · 2024-08-07
>
> **Q1: Evaluation on ScaleVLN.**\
> **A1:** Per your request, we conduct additional experiments on ScaleVLN [49] using the original hyperparameters of ENP (L226-228).
>
> |Models|Training Data of R2R|*val unseen* SR|*val unseen* SPL|*test unseen* SR|*test unseen* SPL|
> |:-:|:-:|:-:|:-:|:-:|:-:|
> |DUET [6]|Original Data [1]|71.52|60.02|69|59|
> |**ENP-DUET (ours)**|Original Data [1]|**74.39**|**60.24**|**71.43**|**60.31**|
> |DUET [6]|ScaleVLN [49]|79|70|77|68|
> |**ENP-DUET (ours)**|ScaleVLN [49]|**80.40**|**70.46**|**79.25**|**69.39**|
>
> As seen, ENP outperforms DUET with the same training data from ScaleVLN, *e.g.*, **1.40**% and **2.25**% SR gains over *val unseen* and *test unseen*, respectively.
>
> **Q2: *"IL/BC is much more friendly and stable than ENP to scale with data."***\
> **A2:** **First**, we have validated the effectiveness of ENP on large-scale training data in **Q1**. We agree that classic behaviour cloning (BC) is straightforward and easy to train for sequential decision-making in VLN. However, as mentioned in many studies [13,16,29], BC suffers from accumulation errors during both training and inference due to covariate shift (L25-35). **Second**, ENP utilizes persistent contrastive divergence (PCD) [73] to train the energy-based models (L182-190). The stability of PCD has been widely demonstrated in various tasks, such as classification [76], generation [23], reinforcement learning [77,78], and natural language processing [ref-1]. **Third**, the training curves provided in the PDF of "global" response (Fig. S3) clearly show that our ENP maintains a stable training process on both the original R2R data (14K) and ScaleVLN (4.9M). We will clarify this in the Experiment of the final version.
>
> [ref-1] Residual energy-based models for text generation. ICLR 2020.
>
> **Q3: *"It is unclear if the method will still benefit larger models ..."***\
> **A3:** It is interesting to investigate ENP on larger models. We noted that NaviLLM [62], which is built updon a large language model (Vicuna-7B [ref-2]), uses the similar pre-training and finetuning paradigm [6] with BC, so ENP can be seamlessly integrated into this framework. Due to the limited time, we incorporate ENP ($I=15$, $\epsilon=1.3$, $Var(\xi)=0.01$) during the finetuning stage. From the following results, we observe that ENP gives **2.43**% and **2.78**% improvements on SR of R2R *val unseen* and SR of REVERIE *val unseen*, respectively. More results will be provided in the Experiment of the final version.
>
> |Models|R2R *val unseen* SR|R2R *val unseen* SPL|REVERIE *val unseen* SR|REVERIE *val unseen* SPL|
> |:-:|:-:|:-:|:-:|:-:|
> |NaviLLM [62]|67|59|42.15|35.68|
> |ENP-NaviLLM|**69.43**|**60.47**|**44.93**|**36.67**|
>
> [ref-2] Instruction tuning with GPT-4.
>
>
> **Q4: More analysis on successful/failure cases.**\
> **A4:** We provide more analysis of ENP *vs.* DUET on success/failure cases in the PDF material of "global" response (Fig. S2). As illustrated in Fig. S2 (a), our agent with ENP successfully reaches the final location, whereas DUET fails due to the accumulation of errors over long paths. In Fig. (b), both our agent and DUET struggle to find the target landmark due to the lack of a specific landmark recognition module like [50,51]. We will incorporate this figure into Fig. 3 in the final revision.
>
> **Q5: *"... in what scenarios an agent trained with ENP shows an advantage."***\
> **A5:** **First**, in Table 3 and 4, ENP achieves consistent performance gains in both *seen* and *unseen* environments across different datasets (R2R-CE [33] and RxR-CE[37]). **Second**, the gains in *unseen* are relatively more significant than that in *seen*. For example, ENP achieves **5.18**% gains of SR on R2R *val seen* (>10m) and **9.35**% gains on *val unseen* (>10m) as follows. **Third**, we further observe that our ENP gives more improvements on challenging cases with long trajectories as follows. ENP demonstrates its capability on R4R [ref-3], where the average path length is about twice that of R2R. The average success rate of ENP exceeds that of BC by **6.40**% on *val seen* (>20m). The performance gain can also significantly reach up to **9.03**% on R4R *val unseen* (>20m).
>
> |Models|R2R *val seen* (<10m)|R2R *val seen* (>10m)|R2R *val unseen* (<10m)|R2R *val unseen* (>10m)|R4R *val seen* (<20m)|R4R *val seen* (>20m)|R4R *val unseen* (<20m)|R4R *val unseen* (>20m)|
> |:-:|:-:|:-:|:-:|:-:|:-:|:-:|:-:|:-:|
> |BC-DUET|76.33|65.50|72.51|46.08|70.88|57.25|52.52|43.71|
> |ENP-DUET|77.69|70.68|74.63|55.43|72.67|63.65|54.91|52.74|
> |$\Delta$|+1.36|+**5.18**|+2.12|+**9.35**|+1.79|+**6.40**|+2.39|+**9.03**|
>
> This indicates that ENP alleviates the cumulative errors in sequential decision-making (L200-210). In the PDF of "global" response, we provide a detailed analysis of success rate across different path lengths (Fig. S1). We will include it in the Appendix of the final version.
>
> [ref-3] Stay on the Path: Instruction Fidelity in Vision-and-Language Navigation. ACL 2019.
>
> **Q6: More explanation about energy-based models (EBMs).**\
> **A6:** Here, we analyze the EBMs from the perspective of sequential prediction [ref-4]. Existing VLN models are based on locally normalized models which optimize the prediction through maximum likelihood at each step. To multigate the exposure bias [ref-4], EBMs define a global energy function at the sequence level, where lower energy values correspond to higher probabilities [21]. Then the normalized probability density can be obtained through dividing by the partition function (Eq. 4). Thus, estimating the partition function is crucial for training EBMs. There is a long history in machine learning [20] for training EBMs. We adopt persistent contrastive divergence [22,73] to sample the partition function and optimize the EBMs (L182-190). We hope it will help you understand more clearly. We will add a detailed background of the EBMs in the Related Work.
>
> [ref-4] Sequence level training with recurrent neural networks. ICLR2016.

---

> ### Comment · Reviewer_Ngjf · 2024-08-12
> **Final Rating**
>
> Thanks to the authors for providing a clear and solid response to my concerns. I am happy with the additional results that justify the data and model scalability and successful/failure cases (questions raised by more than one reviewer), given the limited rebuttal time. I decided to raise the rating from Weak Accept (6) to Accept (7). I sincerely hope that the authors can further extend and incorporate all responses into the final version of their paper.

---

> ### Author Response · Authors · 2024-08-12
>
> Dear Reviewer Ngjf,
>
> Thank you very much for your valuable time and thoughtful review. We are pleased to hear that our rebuttal addressed your concerns, prompting a rise in the rating. We will revise our paper according to your comments and incorporate all suggestions into the final version.
>
> Thanks again,\
> Authors

---

### Official Review · Reviewer_UeCa · 2024-07-14

**Soundness:** 3
**Presentation:** 3
**Contribution:** 3
**Rating:** 6
**Confidence:** 5

**Summary:**

This paper proposes to learn a better Vision-and-Language Navigation agent by jointly optimizing the action and the state. Specifically, they propose an energy-based expert policy, where they adds an explicit objective for navigation exploration and incorporates prior knowledge from the expert demonstrations. Empirical results on multiple VLN benchmarks (e.g., R2R, REVERIE, R2R-CE) demonstrates the effectiveness of the proposed approach.

**Strengths:**

1. The proposed approach is supported with good theoretical foundations, and this paper provides a detailed derivation of the objective, as well as providing explanation on how to place the approach in previous literature with imitation learning approaches.
2. Empirical results on multiple VLN benchmarks demonstrate the effectiveness of the approach, and also demonstrates this approach's generalization to different base VLN agents.
3. Detailed ablations to investigate different components of the approach (e.g., step size, gaussian noise, SGLD loop).

**Weaknesses:**

1. The proposed approaches work pretty well on sota agents like DUET, but it's underinvestigated whether the proposed approach also benefits VLN agents when having large-scale data. Specifically, it's important to have experiments on ScaleVLN or HM3D-AutoVLN.
2. The approach might be sensitive to hyperparameters as suggested in Table 6 and Table 7, which might need additional efforts in hyperparameter tunning when generalized to different environments/agents/tasks.

**Questions:**

Please refer to weakness.

---

> ### Author Rebuttal · Authors · 2024-08-07
>
> **Q1: *"... whether the proposed approach also benefits VLN agents when having large-scale data."***\
> **A1:** Thank you for your comments. The proposed ENP is a general framework to mitigate the accumulation of errors in the sequential decision-making process of VLN. To demonstrate the effectiveness of our ENP on large-scale data, we conduct experiments on ScaleVLN [49], without image augmentation [48] as follows.
>
> |Models|Training Data|R2R *val unseen* SR|R2R *val unseen* SPL|R2R *test unseen* SR|R2R *test unseen* SPL|
> |:-:|:-:|:-:|:-:|:-:|:-:|
> |DUET [6]|Original Data [1]|71.52|60.02|69|59|
> |**ENP-DUET (ours)**|Original Data [1]|**74.39**|**60.24**|**71.43**|**60.31**|
> |DUET [6]|ScaleVLN [49]|79|70|77|68|
> |**ENP-DUET (ours)**|ScaleVLN [49]|**80.40**|**70.46**|**79.25**|**69.39**|
>
> As seen, ENP gives consistent improvement with large-scale training data (*i.e.*, **1.40**% and **2.25**% SR on val unseen and test unseen, respectively). Please note that all the experiments above utilize the original hyperparameters of ENP (L226-228). We will update this result in Table 1 of the final version.
>
> **Q2: *"The approach might be sensitive to hyperparameters as suggested in Table 6 and Table 7 ..."***\
> **A2:** We clarify the robustness of ENP to hyperparameters based on the following aspects. **First**, as shown in Table 6 and Table 7, ENP achieves superior performance with **consistent hyperparameters for the same model across different tasks** (e.g., R2R [1] and R2R-CE [32]). **Second**, ENP is **insensitive to parameter changes within adjacent ranges**. Due to limited space in the paper, we reported only a subset of the results in Tables 6 and 7. For Table 6, we provide detailed results of $\epsilon$ as follows. We find that it has minimal impact on the performance when $\epsilon$ varies between 1.3 and 1.9.
>
> |Models|#Loop|R2R *val unseen* SR|R2R *val unseen* SPL|R2R-CE *val unseen* SR|R2R-CE *val unseen* SPL|
> |:-:|:-:|:-:|:-:|:-:|:-:|
> |ENP-VLNBERT|$\epsilon=1.0$, $Var(\xi)=0.01$|64|**57**|**45**|39|
> |ENP-VLNBERT|$\epsilon=1.1$, $Var(\xi)=0.01$|**65**|**57**|**45**|39|
> |ENP-VLNBERT|$\epsilon=1.3$, $Var(\xi)=0.01$|**65**|**57**|**45**|**40**|
> |ENP-VLNBERT|**$\epsilon=1.5$, $Var(\xi)=0.01$**|**65**|**57**|**45**|**40**|
> |ENP-VLNBERT|$\epsilon=1.7$, $Var(\xi)=0.01$|**65**|**57**|**45**|**40**|
> |ENP-VLNBERT|$\epsilon=1.9$, $Var(\xi)=0.01$|**65**|**57**|44|**40**|
>
> **In addition**, more experimental results related to Table 7 are presented below. Obviously, when the hyperparameter $I$ is between 15 and 19, the performance of ENP-VLNBERT remains stable.
>
> |Models|#Loop|R2R *val unseen* SR|R2R *val unseen* SPL|R2R-CE *val unseen* SR|R2R-CE *val unseen* SPL|
> |:-:|:-:|:-:|:-:|:-:|:-:|
> |ENP-VLNBERT|$I=14$|**65**|56|**45**|39|
> |ENP-VLNBERT|**$I=15$**|**65**|**57**|**45**|**40**|
> |ENP-VLNBERT|$I=16$|**65**|**57**|**45**|**40**|
> |ENP-VLNBERT|$I=17$|**65**|**57**|**45**|**40**|
> |ENP-VLNBERT|$I=18$|**65**|**57**|**45**|**40**|
> |ENP-VLNBERT|$I=19$|**65**|56|44|**40**|
> |ENP-VLNBERT|$I=20$|64|56|44|39|
>
> We will definitely add these detailed tables to the Appendix in the final version.

---

> > ### Comment · Reviewer_UeCa · 2024-08-12
> > **Final Rating**
> >
> > Thanks for addressing my concerns. The additional experiments are very strong. I will raise my rating to weak accept (6).

---

> > > ### Author Response · Authors · 2024-08-12
> > >
> > > Dear Reviewer UeCa,
> > >
> > > Thank you again for your positive feedback and for considering our rebuttal. We will include these experimental results in the final version of our paper. Please do not hesitate to post comments if there are any further questions we can address.
> > >
> > > Thank you,\
> > > Authors

---

### Author Rebuttal · Authors · 2024-08-07

**To all reviewers:**

Thank you very much for your valuable time and suggestive comments. In response, we have meticulously addressed each point raised and provided point-to-point clarifications. Also, the final version of our manuscript will be updated accordingly.

We appreciate the positive feedback from **Reviewer UeCa** and **Ngjf**. Their main concern is about the performance of our ENP on larger-scale datasets, such as ScaleVLN [49]. In our response, we report the results and demonstrate the effectiveness of our ENP on large-scale training data.

We are very grateful for the constructive comments from  **Reviewer dJ3y**. To address your concerns, we provide a detailed and thorough analysis for comparing ENP and BC. In addition, we include **supplementary figures in the PDF of this "global" response** for the following aspects:
1. A comparison of navigation accuracy between ENP and BC across different path lengths and environments, as shown in Figure S1.
2. More success/failure cases of our ENP *vs.* BC in Figure S2 (also mentioned by **Reviewer Ngjf**).
3. The training curves for ENP on R2R and ScaleVLN in Figure S3 (also noted by **Reviewer Ngjf**).

For more details, please refer to our responses to each reviewer. We have strived to address each of your concerns and welcome further discussions and insights.

Sincerely yours,\
Authors

---

### Decision · Program_Chairs · 2024-09-25

**Decision:**

Accept (poster)

**Comment:**

This paper tackles vision-language navigation through a perspective that differs from the traditional behavioral cloning methods. Specifically, the authors propose an energy-based method for joint optimization of the feature-action space. Theoretical motivation for this approach is discussed, and strong results are shown across a number of VLN settings.

  Overall, the reviewers appreciated the strong novel contribution in terms of both theoretical foundations and method which differs from traditional approaches, as well as the empirical results and ablations. A number of joint concerns were raised by a number of reviewers such as the lack of validation on larger-scale VLN settings as well as models, as well as success/failure analysis to show exactly where/how this method improves performance. Other weaknesses such as an apparent sensitivity across the hyper-parameters were mentioned. The authors provided a strong rebuttal with new results on larger-scale settings (ScaleVLN) and models (NaviLLM with a Vicuna-7B model). Importantly, the original hyper-parameters were used, showing both good scaling as well as some hyper-parameter insensitivity. Reviewers were overall satisfied with raised scores based on the rebuttal.

  After considering all of the materials for this paper, I recommend acceptance. It provides an interesting novel approach that is not typical in the community, with strong results, potentially paving the way for new exploration in this area. Further, the method seems applicable to other IL problems where BC is common, and I hope that this method can be explored within those contexts and settings as well. I highly encourage the authors to include the rebuttal results and discussions in the main paper, as I believe it significantly strengthens the paper's claims and impact.